# The Effects of Sedation with Dexmedetomidine–Butorphanol and Anesthesia with Propofol–Isoflurane on Feline Grimace Scale^©^ Scores

**DOI:** 10.3390/ani12212914

**Published:** 2022-10-24

**Authors:** Ryota Watanabe, Beatriz P. Monteiro, Hélène L. M. Ruel, Alice Cheng, Sabrine Marangoni, Paulo V. Steagall

**Affiliations:** 1Department of Clinical Sciences, Faculty of Veterinary Medicine, Université de Montréal, Saint-Hyacinthe, QC J2S 2M2, Canada; 2Department of Veterinary Clinical Sciences, Centre for Animal Health and Welfare, Jockey Club College of Veterinary Medicine and Life Sciences, City University of Hong Kong, Hong Kong, China

**Keywords:** cats, pain assessment, analgesia, sedation, anesthesia, Feline Grimace Scale^©^

## Abstract

**Simple Summary:**

A pain assessment is essential to provide appropriate pain relief. The Feline Grimace Scale^©^ (FGS) is a facial expression-based acute pain scale used in feline medicine; increased FGS scores could indicate acute pain in cats requiring analgesia. However, it is unknown if some sedatives and/or anesthetics can bias pain assessment using the FGS. This study aimed to investigate the effects of sedation with dexmedetomidine-butorphanol followed by anesthesia with propofol-isoflurane on the FGS scores of healthy cats. The cats were video-recorded before and up to 24 h after sedation and general anesthesia. Images collected from the videos were randomized and scored independently by four raters who were masked to the treatments. Dexmedetomidine and butorphanol significantly increased the FGS scores at 20 min after administration. General anesthesia with propofol and isoflurane significantly increased the FGS scores at 0.5 h post-anesthesia, but not after. There were no other statistically significant findings. In conclusion, sedation with dexmedetomidine and butorphanol and general anesthesia with propofol and isoflurane increase FGS scores and may bias clinical pain assessment. Although the effects were short-lived, they should be taken into account during acute pain assessment.

**Abstract:**

This study aimed to evaluate the effects of sedation and anesthesia on Feline Grimace Scale^©^ (FGS) scores. Twelve healthy cats were included in a prospective, blinded and randomized, cross-over study with a 14 day wash-out. Saline or dexmedetomidine-butorphanol (Dex-But) was administered intramuscularly before an anesthetic induction with propofol and maintenance with isoflurane. Saline or atipamezole (Dex-But) was administered at the end of the general anesthesia. Video-filming/image capturing was performed before and up to 24 h post-anesthesia. A total of 125 images were evaluated by four raters blinded to the treatment groups using the FGS (ear position/orbital tightening/muzzle tension/whiskers change/head position; action units (AU); scores 0–2 for each AU). The effects of the sedation/anesthesia were analyzed (*p* < 0.05). The total FGS and each AU scores were significantly higher with Dex-But than with saline 20 min post-sedation. In the saline group, the total FGS, orbital tightening, and whiskers and head position scores were significantly higher than baseline at 0.5 h post-anesthesia. In the Dex-But group, the total FGS and each AU scores were significantly higher after sedation, whereas the orbital tightening scores were significantly higher at 0.5 h post-anesthesia when compared with the baseline. None of the other comparisons between or within the groups was significantly different. The sedation with dexmedetomidine-butorphanol and anesthesia with propofol-isoflurane changed the FGS scores on a short-term basis; consequently, they may bias acute pain assessment.

## 1. Introduction

Perioperative pain assessment is important to decide if analgesics are required for pain relief and to ensure patient welfare [1]. In feline medicine, pain assessment is challenging as the pain behaviors may be subtle in a clinical or hospital setting [2]. Therefore, validated pain scales should be used for appropriate pain recognition [3,4,5]. The Feline Grimace Scale^©^ (FGS) is a facial expression-based pain scale with reported validity and intervention level (i.e., cut-off for the administration of analgesics) [4]. The FGS is applicable for medical and surgical (e.g., soft tissue, orthopedics and dental) pain [4,6], and it can be used for both real-time and image-based evaluation [7]. In addition, the FGS can be reliably used by veterinary students and nurses, and cat caretakers [8,9], highlighting its wide application for feline acute pain assessment.

The assessment of pain might be biased in cats with co-existing conditions or when sedatives or anesthetics have been administered. For example, the presence of upper respiratory tract disease, the patient’s demeanor and the use of ketamine were shown to affect pain scores using tools other than the FGS [10,11]. On the other hand, premedication with intramuscular (IM) administration of acepromazine and buprenorphine did not affect FGS scores [7]. Dexmedetomidine is an agonist of α-2 adrenergic receptors that is commonly used as a sedative agent in feline medicine as it provides dose-dependent sedation, analgesia and muscle relaxation [12]. Dexmedetomidine is often administered for sedation and premedication in combination with butorphanol, an agonist of κ and an antagonist of μ opioid receptors [13]. In feline medicine, it is not known how sedation with dexmedetomidine and butorphanol, and anesthesia with propofol and isoflurane could interfere with acute pain assessment. This information is clinically important as these drugs could produce changes in facial expressions that would be confounded with pain (e.g., increased orbital tightening and lowered head position due to deep sedation/unconsciousness), even if the cat was not painful. In this case, inappropriate administration of analgesics (e.g., opioids) could occur and potentially lead to drug-induced adverse effects; therefore, it is fundamental to understand the effects of sedatives and anesthetics on clinical pain assessments as well as the limitations of a given pain scoring tool such as the FGS.

The objective of this study was to evaluate the effects of sedation with dexmedetomidine and butorphanol and general anesthesia with propofol and isoflurane on the FGS scores in healthy cats. The hypothesis was that the FGS pain scores would be significantly increased after the administration of dexmedetomidine–butorphanol and propofol–isoflurane in healthy cats.

## 2. Materials and Methods

The study was approved by the Institutional Animal Care and Use Committee of the Université de Montréal (protocol 20-Rech-2068) and performed at the *Centre hospitalier universitaire vétérinaire* (CHUV), Faculty of Veterinary Medicine (FMV), Université de Montréal, between July and August 2020, according to the Canadian Council on Animal Care guidelines. The study is reported according to the ARRIVE guidelines [14]. The study design was a prospective, blinded, randomized, cross-over trial with a 14 day wash-out period between the two treatments.

### 2.1. Animals

Twelve healthy domestic short hair cats (6 neutered males and 6 neutered females; 3 (2–7) years old; 4.40 ± 0.67 kg) of the FMV’s teaching cat colony were included. Cats were considered healthy based on a physical examination, history, and vaccination and deworming status. The recruitment was performed by two investigators (PS and BM). For each phase of the cross-over, the cats were admitted the day before the experiment (day 0). Sedation and general anesthesia were performed on day 1, and the cats were returned to their housing facilities on day 2 (Figure 1). The same cats were once again admitted for the second phase of the cross-over after a 14 day wash-out period. During hospitalization, they were housed in stainless steel cages in a cat ward, containing a litter box, bedding, a cardboard box for perching and hiding, and a toy. They had free access to water, but food was withheld for no more than 10 h before the general anesthesia. Cats with fearful behaviors, or any known medical conditions (according to their medical records or detected during the physical examination at admission) were not included. Black-coat cats were also not included, as it has been shown that the action units (AU) of the FGS can be difficult to visualize when using images of these cats [4,6]. The exclusion criteria included any medical condition observed during the study that was unknown to the researchers.

### 2.2. Sedation and Anesthesia

The cats were randomly allocated to either the control or treatment group. The group allocation was determined using a randomization plan generator (www.randomization.com) (accessed on 18 June 2020). In the treatment group, the cats received an IM administration of dexmedetomidine (5 μg/kg; 0.5 mg/mL, Dexdomitor, Zoetis Canada, Kirkland, QC, Canada) and butorphanol (0.2 mg/kg; 10 mg/mL, Dorlex, Intervet Canada, Kirkland, QC, Canada), whereas the cats in the control group received saline using the same volume of administration as the treatment group. Immediately after the administration of the dexmedetomidine–butorphanol, the hair at the area of the cephalic vein was clipped, and a eutectic mixture of local anesthetic cream (EMLA cream lidocaine 2.5% and procaine 2.5% cream, Astra Zeneca, Mississauga, ON, Canada) was applied to the skin and covered with plastic film and an adhesive bandage. Twenty minutes later, a 22-G intravenous (IV) catheter was placed using aseptic technique, connected to an injection port and taped accordingly. Anesthesia was induced with a propofol (10 mg/mL, Propoflo 28, Zoetis, Kirkland, QC, Canada) IV to effect (Figure 1). The cats were intubated with a supraglottic airway device (V-gel^®^, Docsinnovent Ltd., London, UK). The appropriate position of the supraglottic airway device was confirmed with capnography. The anesthesia was maintained with isoflurane (Isoflurane USP, Fresenius Kabi, Toronto, ON, Canada) in oxygen for 30 min using isoflurane concentrations required to prevent swallowing and purposeful movements, and blunt palpebral reflexes. Continuous monitoring during the anesthesia was performed using a multi-parametric monitor (LifeWindow 6000 V veterinary multiparameter monitor; Digicare Animal Health, Boynton Beach, FL, USA) including an electrocardiogram, capnography, pulse oximetry, arterial blood pressure (oscillometric technique) and rectal temperature. The eyes were lubricated with an ocular ointment. Lactated Ringer’s solution was administered intravenously at 3 mL/kg/hour. At the end of the anesthesia, the cats were administered atipamezole (0.05 mg/kg; 5 mg/mL, Antisedan, Zoetis Canada, Kirkland, QC, Canada) in the treatment group or the same volume of saline in the control group. A board-certified veterinary anesthesiologist (PS) was responsible for the general anesthesia of all cats.

### 2.3. Assessment of Sedation

Real-time sedation assessment was performed by one male observer (RW) who was unaware of the treatment group using a dynamic interactive visual analog scale (DIVAS) at the baseline, 20 min after sedation and before the IV catheterization, and at 0.5, 2, 4, 8 and 24 h after the end of the anesthesia. Briefly, the DIVAS was scored on a 100 mm line where the score 0 was considered as no sedation and the score 100 was considered as the deepest possible sedation (e.g., lateral recumbency with no reaction to external stimulation) [15].

### 2.4. Video Recording and Image Capture

Video recordings were performed at the same time points of the DIVAS assessment. The cats were transferred to a specific cage for video recording after the DIVAS scoring and acclimated for 5 min. The video recording was performed using a wide-angle lens camera (GoPro Hero 5, GoPro, Riverside, CA, USA) set between the cage bars at the level of the cats’ eyes. The camera was controlled remotely using a smartphone (iPhone XS, Apple Inc, Cupertino, CA, USA) and a mobile phone application (GoPro Quik, GoPro, Riverside, CA, USA). An electric standing lamp was placed approximately 1 m away from the cage to improve lighting during the video recordings (Watanabe et al., 2020). After acclimation, 3 min videos were recorded for later FGS scoring. The videos were randomized using a random permutation generator and renamed to consecutive numbers by the same individual performing the recordings (RW). The image capture of the cats’ faces was performed by a different investigator (AC) who was not aware of the treatment groups and time point, and was not involved with the image scoring. First, the investigator watched the entire video to ensure it could provide a usable image (i.e., the frontal face of the cat could be visualized in the video). Then, a software (Free Video to JPG Converter, DVDVideoSoft, London, UK) was used to automatically produce still images from each video. The videos were recorded at 60 frames/second and the software was programed to produce two frames (still images) per second. Thereafter, from hundreds of images produced per video, one image was initially selected for each third of the video, followed by the selection of a single image per video. The criteria used for the image selection included: a frontal image; visible AU; no vocalization; no sleeping; no grooming; no yawning. Images in which the cat was leaning onto surfaces (e.g., cage wall or floor) were included as long as the AUs of the contralateral side of the face were visible. The selected image was the one with the highest quality and the one most representative of the facial expressions for that video [4,6]. Finally, the selected image was cropped using Photoshop (Adobe Photoshop CS6 V13.0, San Jose, CA, USA) to include the entire face of the cat and the shoulders. The images were not captured if the cat did not face the camera at any time during the video (i.e., no frontal image).

### 2.5. Image Scoring

A total of 125 images (Figure 2) were scored by four raters: BPM (DVM, Ph.D, resident of an alternate route program of the American College of Animal Welfare, female), PVS (MV, MSc, Ph.D, board-certified by the American College of Veterinary Anesthesia and Analgesia, male), SM (MV, female), and RW (DVM, Ph.D, resident of a program registered with the American/European College of Veterinary Anesthesia and Analgesia, male). The raters were blinded to the treatment and timing of the recording and were supplied with the FGS training manual (https://static-content.springer.com/esm/art%3A10.1038%2Fs41598-019-55693-8/MediaObjects/41598_2019_55693_MOESM1_ESM.pdf (accessed on 1 December 2020)) before the evaluation. The AU ear position, orbital tightening, muzzle tension, whiskers change, and head position were scored as 0 = AU is absent; 1 = moderate appearance of the AU, or uncertainty over its presence or absence; 2 = obvious appearance of the AU; or “not possible to score” if the AU was not clearly visible in the image [4]. The images were scored using an online survey program (SurveyMonkey, Momentive Inc. San Mateo, CA, USA) and divided into two sets (i.e., 63 and 62 images/set). The interval between these two sets was between 24 to 48 h to avoid raters’ fatigue. The scoring was performed between 20 and 23 December 2021. If a cat was in lateral recumbency (e.g., after sedation), the raters scored the AUs based on the half of the cat’s face that was not in contact with the cage surface. Images receiving “not possible to score” for two or more AUs were excluded from the statistical analyses. The total FGS scores were calculated as a ratio (sum of the scores from each AU divided by the maximum possible score based on the number of AUs that were scored).

## 3. Statistical Analyses

The sample size (i.e., the number of images) of this study was determined based on a previous study in which 110 images were included for the development and validation of the FGS [4]. The statistical analyses were performed using the SPSS software (version 27.0 IBM SPSS Statistics, Armonk, NY, USA). The data were tested for normality using the Shapiro–Wilk test. The dose of propofol for the supraglottic airway device placement and time from the end of general anesthesia to extubation were analyzed using a paired t-test and Wilcoxon-signed-rank test, respectively. The scores from the DIVAS (0–100), each AU (0, 1, 2) and total FGS ratios (0–1.0) were compared between the baseline and each time point, and between the groups using linear mixed models for repeated measures. The time and the treatment group, and their interaction were considered as fixed effects. A cat was considered a random effect. An adjustment of the alpha level for each comparison was performed using the Benjamini–Hochberg procedure. Values of *p* < 0.05 were considered statistically significant.

## 4. Results

Two cats (1 male and 1 female) were excluded after the 1st and 2nd phases of the cross-over because of the development of unexpected dysrhythmias during anesthesia and the presence of a superficial eye ulcer, respectively. One female cat was excluded after the image capture because of the development of facial hemi-paralysis, diagnosed after the study by a board-certified veterinary neurologist (HLMR). Therefore, images from 10 cats were available from the 1st phase and images from 9 cats were available from the 2nd phase of the cross-over. The dose of propofol for the anesthetic induction was significantly lower in dexmedetomidine-butorphanol (2.67 ± 0.87 mg/kg) when compared with the control group (5.70 ± 1.52 mg/kg) (*p* < 0.001). The time to extubation (median (range), minutes) was not different between the control (3 (0–5)) and the treatment (3 (1–8)) groups (*p* = 0.83).

The DIVAS scores are shown in Table 1. In the treatment group, the DIVAS scores were significantly higher when compared with the control group at post-sedation and lower when compared with the control group at 0.5 h post-anesthesia. When compared with the baseline, the DIVAS scores were significantly higher at post-sedation in the treatment group and at 0.5 h post-anesthesia in both groups.

The FGS scores are shown in Table 2. The total FGS and each AU scores were significantly higher in the treatment group when compared with the control group after sedation, but not after the general anesthesia (Figure 3).

In the control group, the total FGS, orbital tightening, and whiskers and head position scores were significantly higher at 0.5 h after the end of anesthesia (Figure 4), but not 20 min, when compared with the baseline. In the treatment group, the total FGS and each AU scores were significantly higher after sedation, whereas the orbital tightening scores were significantly higher at 0.5 h after the end of anesthesia, when compared with the baseline (Table 2).

## 5. Discussion

According to the original hypothesis, this prospective, randomized, cross-over study showed that sedation with dexmedetomidine and butorphanol followed by general anesthesia with propofol and isoflurane increased FGS scores at specific time points in healthy, non-painful cats. Therefore, these drugs could affect acute pain assessment in cats using the FGS, as changes in facial expressions or AUs might be related to sedation or post-general anesthesia, and not necessarily pain (Figure 3 and Figure 4). Our results demonstrated that all AUs, and hence, total FGS scores are affected by sedation using dexmedetomidine and butorphanol when compared with a placebo and before sedation for at least 20 min. The effects of general anesthesia on the FGS scores were short-lived (only observed at 0.5 h post-anesthesia) and, apparently, of a smaller magnitude than the effects of sedation (total FGS ratio scores were 0.14 ± 0.04 and 0.51 ± 0.05 after sedation and 0.34 ± 0.05 and 0.23 ± 0.05 after general anesthesia in the saline and the dexmedetomidine-butorphanol groups, respectively).

The doses of sedatives used in the current study (i.e., dexmedetomidine 5 μg/kg and butorphanol 0.2 mg/kg IM) are commonly administered for mild to moderate sedation in our clinical setting. Previous studies using higher doses of dexmedetomidine at 10 μg/kg and butorphanol at 0.4 mg/kg IM found that the onset of sedation was 5 min [13,16], with a peak effect at 15 min post-sedation (Nagore et al. 2012). In our study, the FGS and DIVAS scoring were performed at 20 min after sedation, which should be appropriate to detect the maximum effect of sedation on the FGS scores. Dexmedetomidine, but not butorphanol, was antagonized at the end of the general anesthesia. Considering a duration of effect between 1 and 2 h for butorphanol [17,18], sedation should be minimum at 0.5 h after the end of general anesthesia (approximately 1.5 h after the administration of dexmedetomidine and butorphanol), even if an antagonist of opioid receptors (e.g., naloxone) was not administered. Indeed, only the orbital tightening scores were increased at 0.5 h after the end of anesthesia in the dexmedetomidine-butorphanol group.

In this study, propofol and isoflurane were chosen for general anesthesia as they are commonly used in our institution. The control group required significantly higher doses of propofol (5.7 mg/kg) than the treatment group (2.7 mg/kg). This was not surprising as sedation with dexmedetomidine–butorphanol produces an anesthetic-sparing effect in cats [19]; however, these different doses of propofol for anesthetic induction may have biased FGS scoring. For instance, the total FGS, orbital tightening, and whiskers and head position scores were greater at 0.5 h post-anesthesia in the control group than the baseline values. On the other hand, only the orbital tightening scores were increased at the same time point in the treatment group. Increases in the FGS scores at 0.5 h after the end of the anesthesia could be also associated with residual effects of isoflurane. In general, the hepatic metabolism of isoflurane is low (0.2%), and most of the gas is eliminated via the lungs [20]; therefore, the impact of residual anesthesia with isoflurane on the FGS scores should be minimal. However, in DBA/2, but not CBA mice, short-term anesthesia (10 min) with isoflurane at 2.5% increased the mouse grimace scale scores at 0.5 h post-anesthesia [21]. Hence, it is possible that the effects of isoflurane on grimace scales is breed/genetic-dependent, which could not be verified in this study as only domestic short-hair cats were included. The duration of anesthesia with isoflurane may have influenced the results of our study. In rats, the duration of anesthesia with 2% isoflurane affected the rat grimace scale scores after 10, but not 2 min [22]. In the current study, general anesthesia was maintained for 30 min, and longer or shorter anesthetic periods could potentially influence the FGS scores in a different manner.

Anesthetic recovery and isoflurane requirements during anesthesia may have also affected our findings. Although a study in rats indicated that the duration of anesthetic recovery could influence post-anesthesia behaviors [23], the time to extubation was approximately 3 min in both groups leading to similar anesthetic recoveries. On the other hand, the expired isoflurane concentrations were approximately 0.9% and 1.3% in the treatment and control groups, respectively, which could explain the different results between the groups at 0.5 h post-anesthesia, as the cats were maintained at different depths of anesthesia. As much as this issue can be a limitation of the study, our methodology mimics clinical practice as the anesthetic depth is adjusted according to patient needs and the drugs administered to maintain homeostasis. Additionally, the individuals involved with the FGS and DIVAS scoring were blinded to the treatment groups to minimize any bias caused by different anesthetic depths during the study.

There are some limitations in this study and some have been already discussed. First, the effects of sedation and general anesthesia were tested at the same time. As described above, the treatment group required lower doses of propofol and isoflurane concentrations when compared with the control group, and this might have influenced our results. Second, a supraglottic airway device was used in this study, which requires less induction agent when compared with endotracheal intubation [24]. Therefore, if endotracheal intubation was performed in this study, the propofol requirements could have been higher, which may have changed the FGS scores post-anesthesia. Third, the cats were moved to a specific cage for the video filming which could have affected their behaviors as a response to the new environment. It is unknown if the duration of the acclimation was enough; however, the fact that these cats normally live together and know each other’s scents (pheromones) likely contributed to reducing a possible stress response to the new environment. Fourth, the video recording for the FGS scoring was performed at 0.5 h and then at 2 h post-anesthesia. It is not clear how general anesthesia affects FGS scores before 0.5 h (e.g., 0.25 h) or between these two time points (e.g., 1 h post-anesthesia). Additionally, it is not clear for how long FGS scoring is affected by sedation as the cats were anesthetized shortly after the 20 min post-sedation time point. Fifth, this study only looked at the effects of a single sedation regimen (i.e., dexmedetomidine-butorphanol) and anesthetic protocol. It is not known how different doses and routes of administration, or how each drug administered alone or with other analgesics/anesthetics would have affected our results. In a previous study, the FGS scores were not significantly changed by the administration of IM acepromazine–buprenorphine [7]. Sixth, the present methodology did not investigate the effects of sedation and general anesthesia on other validated pain scoring tools in cats. It is possible that other instruments may also be biased by sedative and anesthetic drugs, as the Glasgow acute pain scale for cats and the UNESP-Botucatu scale for feline pain assessment include the evaluation of some facial expressions [3,5]. As much of the administration of specific sedatives and anesthetics have now been shown to bias acute feline pain assessment using the FGS, the tool has gone through robust validation with appropriate inter- and intra-reliability, and it offers excellent discriminatory ability and a cut-off for the administration of analgesics [4,7]. These studies have included controls with or without the administration of drugs. This study is important as it highlights the potential bias of acute pain scoring tools in the presence of sedation and shortly after general anesthesia.

## 6. Conclusions

Sedation with IM administration of dexmedetomidine and butorphanol, and general anesthesia with propofol and isoflurane may bias acute pain assessments using the FGS in domestic cats. These effects are short-lived, especially after general anesthesia, but should be considered during clinical pain assessment. Veterinary health professionals should be aware of these drug-induced potential effects and that acute pain assessment may not be reliable when dexmedetomidine–butorphanol is administered or in the early phases of anesthetic recovery with propofol–isoflurane. Continuous pain assessment is essential; analgesia can be administered if pain is suspected even if the FGS scores might be biased.

## Data Availability

Data are available from authors upon reasonable request.

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
