# Peer review of "The Effects of Sedation with Dexmedetomidine–Butorphanol and Anesthesia with Propofol–Isoflurane on Feline Grimace Scale© Scores"

_animals, 2022, doi:10.3390/ani12212914_

Round 1

Reviewer 1 Report

Overall, the manuscript report interesting and relevant data about pain scoring in perioperative settings in cats, but several points require re-elaboration, here my detailed comments/suggestions.

In the abstract, you state that 10 cats were used, but in the results session you mention 12 and then 3 excluded, among them one after images scoring (what does this mean? Images included or not?)…this need clarification and consistency throughout the manuscript

Figure 1 is very unclear, the second line seems a prosecution of the first. Please make a new proposal. Post-Ax not explained.

Figure 2: not clear and not logically built. In the same square, cats excluded and videos (first square on the right side)? These are two different "levels", you should differentiate to make the flowchart understandable. Why 3 video? Are they from the same cat? Why are images excluded from the square reporting videos? I have problems to understand this figure based on what you explained in the results session. How many videos did you have per subjects? From the total 151 videos, how was the distribution in the treatment groups?

Lines 155-156: not totally clear how the images were selected (lines 155-156): what are the most representative images? Was the selection done considering what seen in the remaining of the same video? Or just quality-wise the best face images?

Materials and methods: the fact that cats were moved into an instrumented ad hoc cage for video recordings might have affected their arousal status, this must be mentioned/addressed in the discussion.

Table 1: in "Baseline" the control data are repeated twice. This should be corrected.

Table 1 and Figure 3 report the same data. Figure 3 does not add anything new and does not report data variability (range), can be removed.

Table 2: why for scores do you report mean values (SE) (while for DIVAS you reported medians and ragnes)? In the text, unclear that you report about ratio. Same results shown in Table 2 and Figure 4, figure 4 does not bring any additional information.

Figure 6: Both cats belong to the treatment group, right? It would be nice to see a cat of the control group.

Several references are wrong in the text (surely from 21 on…), please check them all and correct.

Author Response

The authors wish to thank the reviewers for their careful review of this manuscript, as well as for the suggestions for improvements. Please find the responses to the reviewers in this document in red italics. Changes to the manuscript are highlighted in yellow.

Reviewer 1 Comments and Suggestions for Authors

Overall, the manuscript report interesting and relevant data about pain scoring in perioperative settings in cats, but several points require re-elaboration, here my detailed comments/suggestions.

Response: Thank you for your reviewing the manuscript.

In the abstract, you state that 10 cats were used, but in the results session you mention 12 and then 3 excluded, among them one after images scoring (what does this mean? Images included or not?)…this need clarification and consistency throughout the manuscript

Response: A total of 12 cats were included in the study. Two cats were excluded during the experimental phase and one cat was excluded during image capture. This has been clarified throughout the text and in Figure 2. 

Figure 1 is very unclear, the second line seems a prosecution of the first. Please make a new proposal. Post-Ax not explained.

Response: Thank you for the suggestion. The second line indicates an enlarged view of the first line. Figure 1 and its captions were edited for improved clarity.

Figure 2: not clear and not logically built. In the same square, cats excluded and videos (first square on the right side)? These are two different "levels", you should differentiate to make the flowchart understandable. Why 3 video? Are they from the same cat? Why are images excluded from the square reporting videos? I have problems to understand this figure based on what you explained in the results session. How many videos did you have per subjects? From the total 151 videos, how was the distribution in the treatment groups?

Response: Thank you for the comments. The figure 2 was modified for better understanding. Within the excluded images, 4 images at post-sedation in the treatment group and 1 image at 0.5 hour post-anesthesia were excluded during the quality assessment. The main reason of exclusion at post-sedation was too deep sedation, which made it not possible to evaluate > 2 AUs.

Lines 155-156: not totally clear how the images were selected (lines 155-156): what are the most representative images? Was the selection done considering what seen in the remaining of the same video? Or just quality-wise the best face images?

Response: Further information was provided to clarify these questions.

 Materials and methods: the fact that cats were moved into an instrumented ad hoc cage for video recordings might have affected their arousal status, this must be mentioned/addressed in the discussion.

Response: Thank you for the comment. This is now discussed as one of the limitations of the study.

Table 1: in "Baseline" the control data are repeated twice. This should be corrected.

Response: Thank you. The formatting of the table was indeed confusing which led to the impression that there were two mentions of the control group for baseline. This has been corrected.  

Table 1 and Figure 3 report the same data. Figure 3 does not add anything new and does not report data variability (range), can be removed.

Response: Figure 3 has been deleted.

Table 2: why for scores do you report mean values (SE) (while for DIVAS you reported medians and ragnes)? In the text, unclear that you report about ratio. Same results shown in Table 2 and Figure 4, figure 4 does not bring any additional information.

Response: DIVAS scores data were not normally distributed and thus reported as median (range). FGS scores data were normally distributed and calculated from the linear models. Thus, these were reported as mean ± SE.

Ratios for the total FGS scores had to be calculated because of the option ‘not possible to score’ when an AU could not be visualized during image assessment. This has been validated in the original study of the FGS (Evangelista et al. 2019) and is explained in section 2.5 of this manuscript.  

Figure 4 has been deleted.

Figure 6: Both cats belong to the treatment group, right? It would be nice to see a cat of the control group.

Response: Thank you. The authors apologize as there was an error in the legend. Further information was added to the captions of figure 6.  

Several references are wrong in the text (surely from 21 on…), please check them all and correct.

Response: The authors apologize for the confusion. There was a problem with reference 1 in the previous version which affected all references. This issue is now resolved.

Reviewer 2 Report

Recognizing and reducing pain is critical in maintaining good animal welfare. This study investigates the affect of sedation with dexmedetomidine-butorphanol followed by anesthesia with propofol-isoflurane on the Feline Grimace Scale© scoring of healthy cats, which plays an important role in feline pain assessment. The research design and analysis are solid, and the data interpretation and discussion is thorough and concise. Well done. The only thing I would suggest it to add some advice for first-line veterinary professionals about how these findings could be applied in the real world. Please find my other minor comments below.

Summary & abstract: For readers who are not familiar with FGS, it would be good if authors could briefly explain the meaning of “increased FGS score” in the summary and abstract, and how might such increase bias clinical practice in a non-technical/layperson language.

Line 76-77: “The hypothesis was that the FGS pain scores would be significantly increased after dexmedetomidine-butorphanol and propofol-isoflurane in healthy cats.” Please provide more background information that supports this hypothesis. Authors mentioned the sedation, analgesia and muscle relaxation effects of those medications, but what is the link between these effects and increased FGS score. There seems to be a missing piece here.

Line 141: Reference 15 does not seem to be relevant to the dynamic interactive visual analog scale used in the study. Please check the reference.

2.5. Image scoring: What is the iter-rater reliability among all 4 raters?

Table 2: Please confirm the p value of whisker changes - control group – 0.5 hour post-anesthesia.

Line 333-334: “In a previous study, FGS scores were not significantly changed by the administration of IM acepromazine-buprenorphine.” Please provide the reference.

It would be good if authors could provide a short discussion about how to apply these findings in the real world. For instance, what is the best time point for an accurate FGS assessment post sedation and/or post anesthesia, presumably using the same medications? Or is there any other behavioral or physiological parameters that can be used to help with the pain assessment considering the FGS is not accurate?

Author Response

The authors wish to thank the reviewers for their careful review of this manuscript, as well as for the suggestions for improvements. Please find the responses to the reviewers in this document in red italics. Changes to the manuscript are highlighted in yellow.

Reviewer 2 - Comments and Suggestions for Authors

Recognizing and reducing pain is critical in maintaining good animal welfare. This study investigates the affect of sedation with dexmedetomidine-butorphanol followed by anesthesia with propofol-isoflurane on the Feline Grimace Scale© scoring of healthy cats, which plays an important role in feline pain assessment. The research design and analysis are solid, and the data interpretation and discussion is thorough and concise. Well done. The only thing I would suggest it to add some advice for first-line veterinary professionals about how these findings could be applied in the real world. Please find my other minor comments below.

Response: Thank you for your overall comments. The potential clinical applicability/importance was added at the conclusion.

 Summary & abstract: For readers who are not familiar with FGS, it would be good if authors could briefly explain the meaning of “increased FGS score” in the summary and abstract, and how might such increase bias clinical practice in a non-technical/layperson language.

Response: Thank you. The information was added to the simple summary.

Line 76-77: “The hypothesis was that the FGS pain scores would be significantly increased after dexmedetomidine-butorphanol and propofol-isoflurane in healthy cats.” Please provide more background information that supports this hypothesis. Authors mentioned the sedation, analgesia and muscle relaxation effects of those medications, but what is the link between these effects and increased FGS score. There seems to be a missing piece here.

Response: Thank you. Further information was added in the introduction to justify our hypothesis.

Line 141: Reference 15 does not seem to be relevant to the dynamic interactive visual analog scale used in the study. Please check the reference.

Response: The authors apologize for the confusion. There was a problem with reference 1 in the previous version which affected all references. This issue is now resolved.

 2.5. Image scoring: What is the iter-rater reliability among all 4 raters?

Response: The inter-rater reliability was assessed for internal control and it was similar to our previous studies.. Since the inter-rater reliability of the FGS has been previously published and that findings herein were similar to these previous publications, such findings were not deemed relevant to be added to this report.

Table 2: Please confirm the p value of whisker changes - control group – 0.5 hour post-anesthesia.

Response: Corrected.

Line 333-334: “In a previous study, FGS scores were not significantly changed by the administration of IM acepromazine-buprenorphine.” Please provide the reference.

Response: The reference has been added.

It would be good if authors could provide a short discussion about how to apply these findings in the real world. For instance, what is the best time point for an accurate FGS assessment post sedation and/or post anesthesia, presumably using the same medications? Or is there any other behavioral or physiological parameters that can be used to help with the pain assessment considering the FGS is not accurate?

Response: Thank you for your suggestion. The potential clinical applicability/importance was added at the conclusion.

Reviewer 3 Report

General comments:

This manuscript represents the results of a study of strong clinical importance to the assessment of pain in a perioperative context, focusing on a popular pain assessment tool based on cat facial features. It draws attention to some of the potential confounds that might be present when applying such pain assessment tools and is a valuable contribution to this field of research.  It is generally well written, however some careful attention is needed to improve the clarity and potential reproducibility of the study, especially regarding the study design, statistical approach and also results being reported.

Abstract:

Ln 19 and ln 28 - Please specify what ‘healthy’ means i.e. does it mean free from any health conditions considered to induce pain?

For greater clarity and context, please provide details regarding all time periods where differences in FGS versus controls were compared and any where no significant between condition differences were identified

Introduction:

Ln 50-56 - This section needs some clarification and rewording – several different pain assessment tools are being referenced although the current wording makes it sound like they are part of the same tool i.e. the ‘Feline Grimace Scale  FGS’ (Evangelista et 2019).  

Sensu the above comments, when referring to FGS elsewhere within this report, please accompany this with the appropriate reference so that it is clear which face-based pain assessment tool this refers to

Ln 58-62 - Reference 10 seems to refer to a study that did use the FGS, therefore is it appropriate to reference here in this way?

Ln 76 - ‘The hypothesis was that the FGS pain scores would be significantly  increased after dexmedetomidine-butorphanol and propofol-isoflurane in healthy cats’. – why? please provide a referenced rationale for this hypothesis. As far as I understand, a lot of the AU changes associated with greater pain are assumed to relate to muscle tension/ contraction rather than relaxation, thus wouldn’t a logical hypothesis be that injection with a muscle relaxant such as Dexmedetomidine would reduce FGS rather than increase it?

Materials and methods:

Please indicate here the breeds of study cats

Ln 109 –‘The order of treatment was evenly distributed between the two study periods’ – I don’t understand what this means in relation to your study design

Ln 155 – ‘Screenshots were obtained using the most representative image that would reflect facial expressions at the selected time point [4,6]).’ – please clarify what is meant here by ‘representative’ and how this differs from a rater simply choosing an image where they felt the cat looked more or less relaxed/sedated etc based on this time point they were being observed within

The study is initially described as a ‘cross over study’ with a ‘14 day wash out period’ but this is not clear within the methods section at all, so I’m confused as to whether all cats received both a control and DEX-BUT treatment or not?

Ln 159 –  please indicate which cats/what time points these 125 images were obtained from – if these were not balanced across time points and cats then this would likely present a confound within the statistical analysis

Ln 191 – ‘each AU and total FGS scores were compared between baseline and each time point, and between groups using linear  mixed models for repeated measures’ – because the study design applied in unclear, I’m left a bit confused about what data are being compared and how -please clarify this

Please state what your response variables were within your models.  

Results:

The figures are a great addition and really help the reader to visualise the relevance of the findings  

Some of the results are hard to properly interpret due to aforementioned lack of clarity over design/analytical approach

Line  194 of the methods it states that ‘The best structures  of the covariance (autocorrelation 1) were assessed using information criteria that measured the relative fit of a competing covariance model.’ – the outcome of this and how it was applied in the interpretation and presentation of the results in table 2 is missing

Please provide the full reports for the models (i.e. some indication of effect size, confidence intervals, standard error, df etc) from which interferences are being made - p values on their own are not very informative and it is not clear how the values reported in table 2 related to the specific statistical tests that were undertaken

Discussion:

Some important limitations are highlighted and discussed. Given the findings, it might be useful to suggest how they should be applied/considered clinically during pain scoring – what are the practical take home messages here? Perhaps to not only rely on FGS tools but also apply those that incorporate a range of other measures? And/Or to discount FGS scores during certain time periods?

The study analysed data from what look to be only 9 cats (please clarify this in the manuscript somewhere) – where statistical tests were performed in a between subjects design (I can’t ascertain if this was the case or not at present) then essentially are some comparisons being made between contributions from presumably populations of only 4 and 5 cats? If this is correct then this should be discussed in terms of its substantial limitations

Author Response

The authors wish to thank the reviewers for their careful review of this manuscript, as well as for the suggestions for improvements. Please find the responses to the reviewers in this document in red italics. Changes to the manuscript are highlighted in yellow.

Reviewer 3 - Comments and Suggestions for Authors

This manuscript represents the results of a study of strong clinical importance to the assessment of pain in a perioperative context, focusing on a popular pain assessment tool based on cat facial features. It draws attention to some of the potential confounds that might be present when applying such pain assessment tools and is a valuable contribution to this field of research.  It is generally well written, however some careful attention is needed to improve the clarity and potential reproducibility of the study, especially regarding the study design, statistical approach and also results being reported.

Response: Thank you for your comments.

Abstract:

Ln 19 and ln 28 - Please specify what ‘healthy’ means i.e. does it mean free from any health conditions considered to induce pain?

Response: Correct. But the authors do not believe that ‘healthy’ needs to be defined in an abstract.

For greater clarity and context, please provide details regarding all time periods where differences in FGS versus controls were compared and any where no significant between condition differences were identified.

Response: Thank you for the comment. Additional information was added

Introduction:

Ln 50-56 - This section needs some clarification and rewording – several different pain assessment tools are being referenced although the current wording makes it sound like they are part of the same tool i.e. the ‘Feline Grimace Scale  FGS’ (Evangelista et 2019).

Sensu the above comments, when referring to FGS elsewhere within this report, please accompany this with the appropriate reference so that it is clear which face-based pain assessment tool this refers to

Response: The authors apologize for the confusion. There was a problem with reference 1 in the previous version which affected all references. This issue is now resolved.

Ln 58-62 - Reference 10 seems to refer to a study that did use the FGS, therefore is it appropriate to reference here in this way?

Response: The authors apologize for the confusion. There was a problem with reference 1 in the previous version which affected all references. This issue is now resolved.

Ln 76 - ‘The hypothesis was that the FGS pain scores would be significantly  increased after dexmedetomidine-butorphanol and propofol-isoflurane in healthy cats’. – why? please provide a referenced rationale for this hypothesis. As far as I understand, a lot of the AU changes associated with greater pain are assumed to relate to muscle tension/ contraction rather than relaxation, thus wouldn’t a logical hypothesis be that injection with a muscle relaxant such as Dexmedetomidine would reduce FGS rather than increase it?

Response: Thank you. Further information was added in the introduction to justify our hypothesis.

Materials and methods:

Please indicate here the breeds of study cats

Response: The information was added.

Ln 109 –‘The order of treatment was evenly distributed between the two study periods’ – I don’t understand what this means in relation to your study design.

Response: This sentence was deleted to avoid confusion.

Ln 155 – ‘Screenshots were obtained using the most representative image that would reflect facial expressions at the selected time point [4,6]).’ – please clarify what is meant here by ‘representative’ and how this differs from a rater simply choosing an image where they felt the cat looked more or less relaxed/sedated etc based on this time point they were being observed within

Response: Thank you for the comment. Additional information was added here as also requested by other reviewers.

The study is initially described as a ‘cross over study’ with a ‘14 day wash out period’ but this is not clear within the methods section at all, so I’m confused as to whether all cats received both a control and DEX-BUT treatment or not?

Response: Thank you. Further information was added in the section 2.1.

Ln 159 –  please indicate which cats/what time points these 125 images were obtained from – if these were not balanced across time points and cats then this would likely present a confound within the statistical analysis

Response: The distribution of each treatment in each time point was added to figure 2.

Ln 191 – ‘each AU and total FGS scores were compared between baseline and each time point, and between groups using linear  mixed models for repeated measures’ – because the study design applied in unclear, I’m left a bit confused about what data are being compared and how -please clarify this

Please state what your response variables were within your models.  

Response: Further information was added.

 Results:

The figures are a great addition and really help the reader to visualise the relevance of the findings  

Some of the results are hard to properly interpret due to aforementioned lack of clarity over design/analytical approach

Response: Thank you. Hopefully the study methodology is clearer after the present revisions.

Line  194 of the methods it states that ‘The best structures  of the covariance (autocorrelation 1) were assessed using information criteria that measured the relative fit of a competing covariance model.’ – the outcome of this and how it was applied in the interpretation and presentation of the results in table 2 is missing

Response: This information was deleted from the text to avoid confusion as it does not add to the findings.

When analyzing data with linear models, several models using different structures of covariance are calculated. The ‘Akaike information criterion (AIC)’ is calculated for each of the different models and the one yielding the lowest AIC is interpreted as the best model which is then chosen.

This type of information is not normally reported in the scientific literature as it is simply a step of the data analysis which helps with the choice of the most appropriate model.

Please provide the full reports for the models (i.e. some indication of effect size, confidence intervals, standard error, df etc) from which interferences are being made - p values on their own are not very informative and it is not clear how the values reported in table 2 related to the specific statistical tests that were undertaken

 Response:  Table 2 provides estimate for the central tendency and data variability (mean ± SE) for the scores for each AU and the total FGS ratio scores. Additional information was added to the legend.

Discussion:

Some important limitations are highlighted and discussed. Given the findings, it might be useful to suggest how they should be applied/considered clinically during pain scoring – what are the practical take home messages here? Perhaps to not only rely on FGS tools but also apply those that incorporate a range of other measures? And/Or to discount FGS scores during certain time periods?

Response: Thank you for the comment. Such information was added to the conclusions.

The study analysed data from what look to be only 9 cats (please clarify this in the manuscript somewhere) – where statistical tests were performed in a between subjects design (I can’t ascertain if this was the case or not at present) then essentially are some comparisons being made between contributions from presumably populations of only 4 and 5 cats? If this is correct then this should be discussed in terms of its substantial limitations

Response: Thank you for the comment. Clarifications regarding the study methodology, results and the study’s limitations were added throughout the text.

Since this study was cross-over study, most of the comparisons were done with a population of 9-10 cats/group.

Reviewer 4 Report

This manuscript addresses an interesting information in feline pain-assessment and was very well written and discussed. I have some suggestions which I hope are helpful.

-          In the abstract, is described the involvement of 10 cats in the study, whereas in the Material and Methods section the number informed is 12. I think is better to consider 12, since in the Results the exclusion of the two cats is well explained and justified.

-          Table 1 and Figure 3 appear to present the same information.  As information is provided with more accurate detail in a table than in a figure, my suggestion is to remove figure 3

-           Reference 1 - was not included in the references list

Author Response

The authors wish to thank the reviewers for their careful review of this manuscript, as well as for the suggestions for improvements. Please find the responses to the reviewers in this document in red italics. Changes to the manuscript are highlighted in yellow.

Reviewer 4 - Comments and Suggestions for Authors

This manuscript addresses an interesting information in feline pain-assessment and was very well written and discussed. I have some suggestions which I hope are helpful.

-          In the abstract, is described the involvement of 10 cats in the study, whereas in the Material and Methods section the number informed is 12. I think is better to consider 12, since in the Results the exclusion of the two cats is well explained and justified.

Response: Thank you. This was corrected.

-          Table 1 and Figure 3 appear to present the same information.  As information is provided with more accurate detail in a table than in a figure, my suggestion is to remove figure 3.

Response: Thank you for your suggestion. Figures 3 and 4 were deleted as per this and another reviewer’s suggestion.

-           Reference 1 - was not included in the references list

Response: The authors apologize for the confusion. There was a problem with reference 1 in the previous version which affected all references. This issue is now resolved.